# VISUALIZING THE EMERGENCE OF PRIMITIVE INTER­ACTIONS DURING THE TRAINING OF DNNS

## ABSTRACT

Although the learning of deep neural networks (DNNs) is widely believed to be a fitting process without an explicit symbolic structure, previous studies have dis­covered (Ren et al., 2023a; Li & Zhang, 2023b) and proven (Ren et al., 2023c) that well-trained DNNs usually encode sparse interactions, which can be considered as primitives of the inference. In this study, we redefine the interaction on principal feature components in intermediate-layer features, which significantly simplifies the interaction and enables us to explore the dynamics of interactions through­out the learning of the DNN. Specifically, we visualize how new interactions are gradually learned and how previously learned interactions are gradually forgotten during the training process. We categorize all interactions into five distinct groups (*reliable, withdrawing, forgetting, betraying*, and *fluctuating interactions*), which provides a novel perspective for understanding the learning process of DNNs.

## 1 INTRODUCTION

Explainable artificial intelligence (XAI) has received increasing attention in recent years. A funda­mental challenge in the realm of XAI is to identify primitives that reflect the inference logic behind the output score of deep neural networks (DNNs). Analogous to taking semantic concepts as prim­itives in human cognition, previous researchers hoped to examine whether a DNN also encoded some kind of concepts (Zhou et al., 2016; Kim et al., 2018). However, up to now, there is still no universally accepted formulation of semantic concepts in DNNs, because it involves joint issues in cognitive science and mathematics. In recent research, Ren et al. (2023c) proved that under some common conditions,[1] a well-trained DNN usually encoded a relatively small number of interactions among input variables (*e.g.* pixels and words). Thus, they claimed these interactions could be con­sidered as concepts encoded by the DNN. For example, Figure 1(a) shows a DNN may encode an interaction between two eye patches and a mouth patch, and this interaction makes a numerical util­ity on the classification score of the cat category. Masking any of the three patches will invalidate this interaction and remove its utility from the output.

Although we still doubt whether the interactions in (Ren et al., 2023c) can precisely represent con­cepts in human cognition, a series of studies have indicated that interactions can still serve as primi­tive inference patterns used by a DNN, to some extent. (1) Ren et al. (2023c) proved that even when input variables in an input sample were arbitrarily masked, it was always possible to use a small number of interactions to replicate the output score of the DNN. (2) Li & Zhang (2023b) demon­strated the generalization ability of interactions in classification tasks. (3) Besides, (Cheng et al., 2021) discovered that in image classification, each interaction between image patches potentially corresponded to a specific visual concept.

Therefore, in this study, we adopt the above interactions as the primitives for the output score of a DNN. In this way, the learning dynamics of a DNN can be roughly explained as the gradual emer­gence of discriminative interactions and the progressive forgetting of incorrect interactions through­out the training process.

Thus, our goal is to visualize the changes in primitive interactions during the learning process. How­ever, previous research defined interactions on raw input variables, and the high dimension of inputs boosted the computational complexity. Besides, some interactions between raw input variables are

---

[1]Please see Appendix A for details.

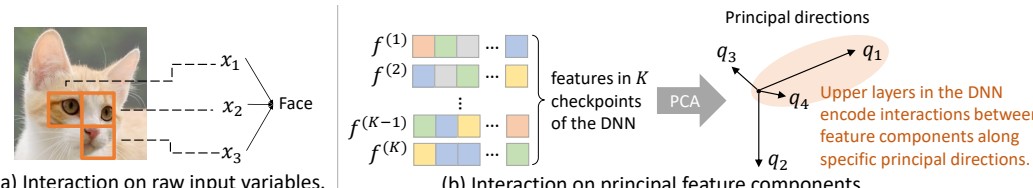

Figure 1: In this study, we extend the interaction based on (a) raw input variables to interactions based on (b) principal feature components.

noisy patterns without a clear relation with the classification task. In order to provide a more concise explanation of the DNN, we redefine the interaction on top-ranked principal feature components in intermediate layers. As Figure 1 (b) shows, we consider the top-ranked principal feature components as basic "input variables" for interactions. We empirically verify that the proposed interactions on feature components are much sparser and more informative for classification than interactions between raw input variables (see Figure 3). The sparsity of the interaction-based explanation suggests a high likelihood of representing the essential inference logic of a DNN, considering Occam's Razor.

In this way, the entire learning process of a DNN can be explained by the emergence of new interactions between feature components and the forgetting of old interactions. Thus, we visualize the emergence and forgetting of these interactions to analyze the learning dynamics of DNNs. We find that interactions encoded in a DNN can be categorized into five groups, namely *reliable, withdrawing, forgetting, betraying*, and *fluctuating* interactions. We further analyze the complexity and number of interactions in each group, which provides new insights into our understanding of the learning of different DNNs.

## 2  RELATED WORKS

**Explaining the inference of a DNN.** Zeiler & Fergus (2014); Simonyan & Zisserman (2014); Dosovitskiy & Brox (2016) directly visualized the receptive fields of intermediate-layer features in the DNN. On the other hand, Ribeiro et al. (2016); Zhou et al. (2016); Selvaraju et al. (2017); Sundararajan et al. (2017); Lundberg & Lee (2017) estimated the attributions or importance of input variables for inference. To explain a DNN as a symbolic system, Che et al. (2016); Frosst & Hinton (2017); Wu et al. (2018); Shih et al. (2019) distilled knowledge from a DNN to symbolic models to explain the inference logic of a DNN. However, these studies lacked sufficient mathematical support to guarantee their explanations really reflect true primitives encoded by a DNN.

**Quantifying interactions in a DNN.** Compared with the above distillation-based symbolic explanation, defining and quantifying interactions within a DNN has emerged as a more straightforward approach to explaining primitives encoded by a model (Murdoch et al., 2018; Singh et al., 2019; Jin et al., 2020; Sundararajan et al., 2020; Janizek et al., 2021; Tsai et al., 2022). Recent works by Ren et al. (2023c); Li & Zhang (2023b) have demonstrated that, under some common conditions,[1] a well-trained DNN tended to encode a small number of interactions for inference. In this context, we propose to define interactions based on principal feature components, and we find that such interactions yield a much simpler explanation for a DNN.

## 3  PRIMITIVE INTERACTIONS IN DNNS

### 3.1  PRELIMINARY: INTERACTIONS

Exploring the primitives of the inference logic constitutes a central concern in explaining a black-box system, just like explaining a set of cognitive concepts in the human brain. To this end, there is not any solid proof of whether a DNN encodes concepts that align with human cognition. Despite issues in cognitive science, Ren et al. (2023c); Li & Zhang (2023b) have provided sufficient mathematical evidence for us to consider interactions as some kind of inference primitives behind the output score of the DNN. Given a trained DNN $v$, let $x \in \mathbb{R}^n$ denote an input sample with $n$ input variables (*e.g.,* an image with $n$ image patches, a sentence with $n$ words, and an intermediate-layer feature with $n$ principal components). Let $N = \{1, 2, \ldots, n\}$ denote the set of indices for these input variables.

The DNN's output is denoted by $v(x) \in \mathbb{R}$.[2] Li & Zhang (2023a) have defined the following two types of interactions, *i.e.*, the AND interaction and the OR interaction.

**Definition 1-$\alpha$ (AND interaction).** *Let us use AND interactions to explain a function $v_{and}(x)$. For a specific subset $S \subseteq N$ of input variables, the utility $I_{and}(S|x)$ of the AND interaction within $S$ is defined as follows. In particular, $I_{and}(\emptyset|x) = v_{and}(x_\emptyset)$.*

$$I_{and}(S|x) = \sum\nolimits_{L \subseteq S} (-1)^{|S|-|L|} v_{and}(x_L) \tag{1}$$

*where $| \cdot |$ denotes the cardinality of a set. For each subset $L \subseteq S$, $x_L$ denotes a masked input sample, in which the variables in $N \setminus L$ are masked.*[3] *Then, $v_{and}(x_L) \in \mathbb{R}$ denotes the output on the masked input. $v_{and}(x_\emptyset)$ denotes the output when we mask all input variables in $N$. Accordingly, we obtain $v_{and}(x_N) \equiv v_{and}(x)$.*

**Definition 1-$\beta$ (OR interaction).** *Let us use OR interactions to explain a function $v_{or}(x)$. For a specific subset $S \subseteq N$ of input variables, the utility $I_{or}(S|x)$ of the OR interaction within $S$ is defined as follows. In particular, $I_{or}(\emptyset|x) = v_{or}(x_\emptyset)$.*

$$I_{or}(S|x) = -\sum\nolimits_{L \subseteq S} (-1)^{|S|-|L|} v_{or}(x_{N \setminus L}) \tag{2}$$

The AND interaction was first proposed by Harsanyi (1959) as the Harsanyi dividend. The OR interaction can essentially be regarded as a specialized form of the AND interaction. This is achieved by considering original variable values as masked states and taking the masked states (baseline values) as conventional values of variables (please refer to Appendix B for details).

Figure 1 shows an interaction in an image of a cat's face, where a function $v_{and}$ may encode the AND interaction within $S = \{x_1, x_2, x_3\}$. The co-appearance of these three patches triggers the AND interaction, thus contributing a numerical utility $I_{and}(S|x)$ to the output $v_{and}(x)$. Conversely, if any patch in $S$ is masked, this AND interaction will be invalidated, and its utility $I_{and}(S|x)$ will be removed from the output, *i.e.*, resulting in $I_{and}(S|x) = 0$. The positive (or negative) utility of the interaction indicates the AND interaction among variables in $S$ will increase (or decrease) the output $v_{and}(x)$. Similarly, the OR interaction represents the OR relationship between variables, *e.g.*, the existence of any patch in $\{x_1, x_2\}$ indicates the eye of the cat. Thus, another function $v_{or}$ may encode an OR interaction in $\{x_1, x_2\}$, and it has a utility $I_{or}(\{x_1, x_2\}|x)$ to the output $v_{or}(x)$.

Li & Zhang (2023a) have proposed a method to extract both AND interactions and OR interactions from a DNN. They disentangled the output score $v(x_L)$ of the DNN into two components $v_{and}(x_L) = 0.5v(x_L) + \gamma_L$ and $v_{or}(x_L) = 0.5v(x_L) - \gamma_L$ with learnable parameters $\{\gamma_L\}$. In this way, it is further proven that the DNN output $v(x)$ can be decomposed as the sum of utilities of all interactions.

$$v(x) = v_{and}(x) + v_{or}(x) = \sum\nolimits_{S \subseteq N} I_{and}(S|x) + \sum\nolimits_{S \subseteq N} I_{or}(S|x) \tag{3}$$

where the overall DNN output $v(x)$ is disentangled into the output score $v_{and}(x)$ for AND interactions $I_{and}(S|x)$ and the output score $v_{or}(x)$ for OR interactions $I_{or}(S|x)$. The parameters $\{\gamma_L\}$ are determined by minimizing $\sum_{S \subseteq N}[|I_{and}(S|x)| + |I_{or}(S|x)|]$.

**Considering interactions as primitives of the inference.** The above interactions can be considered as primitives of the inference logic of the model, because of the following three properties:

• *Sparsity.* Ren et al. (2023c) have proved that under three common conditions,[1] a DNN usually encodes very sparse interactions. Although there can be up to $2^n$ different subsets $S \subseteq N$, they prove that only a few subsets of variables exhibit considerable interaction utility (*i.e.*, $|I_{and}(S|x)|$ or $|I_{or}(S|x)|$ is large) for the DNN output. All other subsets have almost zero interaction utility. Let $\Omega_{salient}$ denote the set of subsets $S$ that have a large utility (large value of $|I_{and}(S|x)|$ or $|I_{or}(S|x)|$).

• *Universal approximating.* Li & Zhang (2023a) have proved that a small number of interactions in $\Omega_{salient}$ are already powerful enough to approximate model outputs on $2^n$ randomly masked inputs.

---

[2] There are various settings for $v(x)$ when the DNN has multiple output dimensions. For example, in multi-category classification tasks, $v(x)$ is usually defined as $v(x) = \log \frac{p(y=y^*|x)}{1-p(y=y^*|x)}$ by following (Deng et al., 2022), where $y^*$ denotes the ground-truth label of the input $x$.

[3] In practice, people usually mask input variables in $N \setminus L$ using baseline values $\{b_i\}$ (also called reference values) (Ancona et al., 2019; Covert et al., 2020) to replace the original values in these input variables, *i.e.*, setting $x_i = b_i$ if $i \in N \setminus L$.

**Theorem 1.** *Given an input sample $x$, when we arbitrarily mask the variables in $x$ to obtain the masked inputs $x_S$ w.r.t. a randomly subset $S \subseteq N$, the DNN output on the input $v(x_S)$ can be accurately mimicked by the sum of interactions,* i.e., $\forall S \subseteq N, v(x_S) = v(x_\emptyset) + \sum_{\emptyset \neq L \subseteq S} I_{and}(L|x) + \sum_{L \cap S \neq \emptyset} I_{or}(L|x) \approx v(x_\emptyset) + \sum_{\emptyset \neq L \subseteq S, L \in \Omega_{salient}} I_{and}(L|x) + \sum_{L \cap S \neq \emptyset, L \in \Omega_{salient}} I_{or}(L|x)$.

• *Generalization power.* Li & Zhang (2023b) have discovered the generalization ability of interactions. That is, people can extract a common set of interactions from different (but similar) inputs or different models, and these interactions are discriminative for classification.

### 3.2  PRIMITIVE INTERACTIONS ON FEATURES

Previous studies commonly regarded raw pixels/words/3D points in an input image/sentence/3D point cloud as the basic input variables of interactions. However, this approach face s challenges when the input is high-dimensional, because it leads to a large number of interactions[4] and also boosts the computational cost. To circumvent these issues, we introduce a new perspective: we redefine interactions on the top-ranked principal feature components in an intermediate layer of a DNN. We discover that compared to interactions defined on raw input variables, defining interactions on feature components can further enhance the sparsity of interactions.

Let us train a DNN, and collect the DNN trained after $K$ different checkpoints (epochs). Given an input sample, we extract the feature from a certain intermediate layer of the DNNs at these $K$ checkpoints, denoted by $f^{(1)}, f^{(2)}, \ldots, f^{(K)} \in \mathbb{R}^m$. Subsequently, we conduct principal component analysis (PCA) on the $K$ features to compute the top $r$ principal directions (eigenvectors) $q_1, q_2, \ldots, q_r \in \mathbb{R}^m$ corresponding to the largest $r$ eigenvalues. In this way, we extract feature components along the top $r$ principal directions, so as to use these feature components as basic "input variables" to define the interaction. Specifically, for the feature $f^{(k)}$ extracted after $k$ epochs, we can decompose the intermediate-layer feature $f^{(k)}$ into the following $(r + 2)$ feature components.

$$f^{(k)} = \sum_{i \in N_{\text{feature}}} f_i + \bar{f} + \epsilon \tag{4}$$

where $N_{\text{feature}} = \{1, 2, \ldots, r\}$ denotes the indices of the top $r$ principal feature components. $f_i = q_i q_i^T (f^{(k)} - \bar{f}) \in \mathbb{R}^m$ represents the $i$-th principal feature component. $\bar{f} = \sum_{k=1}^{K} f^{(k)}/K$ denotes the average feature during the learning process. $\epsilon = f^{(k)} - \bar{f} - \sum_{i \in N_{\text{feature}}} f_i$ is referred to as the overall effect of all the remaining $m - r$ feature components in $f^{(k)}$.

In this way, if we consider $\bar{f} + \epsilon$ as a constant background, we can regard the $r$ feature components in $f^{(k)}$ as the variables involved in interactions. *I.e.,* each interaction $S \subseteq N_{\text{feature}}$ represents the collaborative relationship between feature components in $S$. Here, because $f^{(k)}$ can be extracted from any epoch, we ignore the superscript $(k)$. Then, for a subset $L \subseteq N_{\text{feature}}$, $f_L$ represents the masked feature when we mask feature components in $N_{\text{feature}} \setminus L$,[5] *i.e.,* $f_L = \sum_{i \in L} f_i + \sum_{i \in N_{\text{feature}}, i \notin L} b_i + \bar{f}$. We use $b_i \overset{\text{def}}{=} q_i q_i^T (f|_{\mathbb{E}[x]} - \bar{f})$ to represent the masked state (or namely *the baseline value*) of the $i$-th feature component. $f|_{\mathbb{E}[x]}$ denotes the feature when the average value $\mathbb{E}[x]$ of all input samples in the training set is fed to the model. $b_i$ represents the $i$-th feature component in the feature $f|_{\mathbb{E}[x]}$.

The DNN output $v(x)$ can be also regarded as a function of the feature $f$, *i.e.,* $v(x) = g(f)$, where $g(\cdot)$ denotes the subsequent layers built upon the feature $f$. $g(f_L)$ denotes the DNN output on the masked feature. Thus, we can directly use Eq. (1) and Eq. (2) to compute the utility of interactions $I_{\text{and}}(S|f)$ and $I_{\text{or}}(S|f)$ on feature components by replacing $v(x_L)$ with $g(f_L)$.

**How many principal feature components are needed as input variables?** To ensure the reliability of the extracted interactions, it is crucial to examine whether the top $r$ feature components are sufficiently significant to represent most signals in $f$. Therefore, we conducted experiments to visualize the significance of feature components along all directions (eigenvectors) in the DNN. For each DNN, we fed an input sample $x$ to the DNNs trained after $K$ different epochs, and extracted $K$ feature vectors $f^{(1)}, \ldots, f^{(K)}$ from these DNNs. Figure 2 shows the eigenvalues obtained by applying PCA to $K$ feature vectors. We found that in most DNNs, the top 10 eigenvalues were significantly larger than the rest. This observation suggested that the feature components along the top 10 directions could well represent most signals within the features.

---

[4]Nevertheless, the number of salient interactions with considerable utilities is still significantly lower than the exponential number of all potential $2^{n+1}$ interactions.

[5]Given that the components in $\epsilon$ in Eq. (4) are usually very small (see Figure 2), we ignore these components.

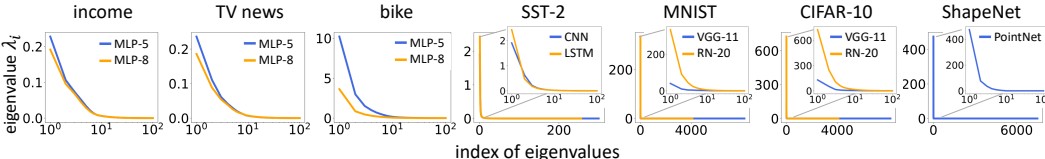

Figure 2: Significance (eigenvalue $\lambda_i$) of feature components in a descending order.

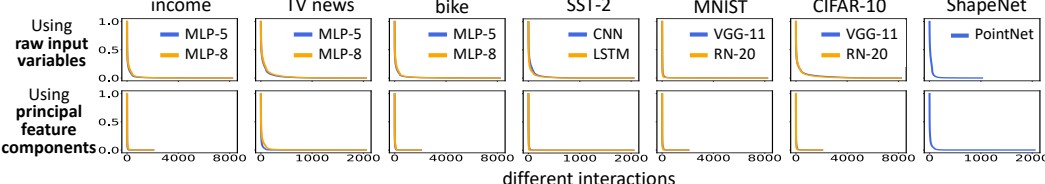

Figure 3: Comparison of sparsity of interactions. Relative strength of interactions on raw input variables and that of interactions on principal feature components. For clarity, AND and OR interactions were put together and sorted in descending order of relative strength. Using principal feature components significantly enhanced the sparsity of interactions.

*Experimental settings.* We trained a 5-layer MLP (Ren et al., 2023b) (namely *MLP-5*) and an 8-layer MLP (Ren et al., 2023b) (namely *MLP-8*) on three UCI tabular datasets (Dua & Graff, 2017), including the census income (namely *income*), TV News channel commercial detection (namely *TV news*), and bike sharing (namely *bike*) datasets. We also followed (Li & Zhang, 2023b) to train a CNN and a three-layer unidirectional LSTM model on the SST-2 dataset (Socher et al., 2013). Besides, we trained VGG-11 (Simonyan & Zisserman, 2014) and ResNet-20 (He et al., 2016) (namely *RN-20*) on the MNIST (LeCun et al., 1998) and CIFAR-10 (Krizhevsky, 2012) datasets, and trained PointNet (Charles et al., 2017) on the ShapeNet (Yi et al., 2016) dataset. For the MLPs, CNN, and LSTM models, we selected output features of the second fully-connected/convolutional/LSTM layer for PCA. For VGG-11, we selected output features of the `conv2_1` layer, and for RN-20, we selected output features of the `conv3_6` layer. For the PointNet, we selected the output features of the input transform layer. We generated different masked samples on each training sample in tabular datasets to generate features on different masked samples to perform PCA. Given each input sample, all of its features collected from different epochs were used for PCA.

**Cost of computing interactions between feature components.** Compared to interactions on raw input variables, interactions defined on feature components present a much smaller computational cost. For the input $x \in \mathbb{R}^n$, the computational cost of interactions on the $n$ input variables in $x$ is $2^n$. When we define interactions on top $r$ feature components ($r \ll m$ in most cases), the computational cost of interactions is reduced to $2^r$, which is much less than $2^n$.

### 3.2.1 SPARSITY OF INTERACTIONS

If the network output on an input sample can always be explained by a small set of interactions, no matter how we randomly mask the input, then the principle of Occam's Razor suggests that we can consider such interactions as primitive inference patterns encoded by the DNN. Otherwise, if a large number of interactions are required to explain the DNN, then these interactions are less likely to reflect the essence of the inference logic used by the DNN. To this end, Ren et al. (2023c) have proven the sparsity of AND interactions on raw input variables under simplifying assumptions, but these assumptions are difficult to examine in real DNNs. Besides, we also extend AND interactions to OR interactions. Therefore, we still need to verify the sparsity of interactions on feature components.

We conducted experiments to compare the sparsity of interactions on feature components with the sparsity of interactions on raw input variables. In order to extract interactions on raw input variables, we followed the experimental settings in (Ren et al., 2023b) to divide each input image in the MNIST and CIFAR-10 datasets into $7 \times 7$ and $8 \times 8$ image regions, respectively. We randomly sampled twelve image regions as twelve input variables to compute interactions. For the ShapeNet dataset, we computed interactions using the manually annotated parts provided by (Li & Zhang, 2023b) as input variables. To compute interactions on the top-ranked principal feature components, we followed the "*Experimental setting*" paragraph to compute principal feature components. Then, we used the top $r = 10$ principal feature components to compute interactions. For simplicity,

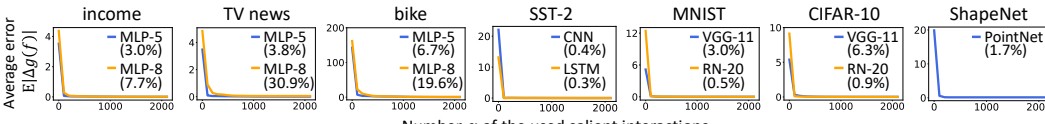

Figure 4: Average error when we use different numbers of interactions to approximate the DNN output. The bracket shows the minimum ratio $\hat{\alpha}/2^{r+1}$ of the most salient interactions (*i.e.*, the top $\hat{\alpha}$ interactions) among all $2^{(r+1)}$ interactions that satisfy $|g(f) - \hat{g}_\alpha(f)|/|g(f)| \le 0.1$.

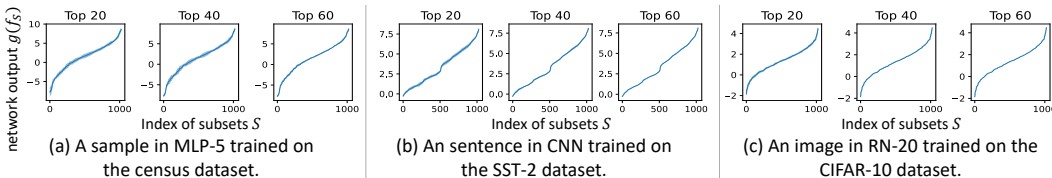

Figure 5: Average error of using salient interaction to approximate network outputs $g(f_S)$ on different masked features. Outputs $g(f_S)$ *w.r.t.* different subsets $S$ were sorted in an ascending order. The shaded area represents the approximation error when using the sum of interactions to match the real network output $g(f_S)$. We averaged the error over neighboring 50 subsets $S$.

we concatenated the strength $|I_{\text{and}}(S|x)|$ of $2^r$ AND interactions and the strength $|I_{\text{or}}(S|x)|$ of $2^r$ OR interactions to construct a $2^{r+1}$-dimensional vector $\boldsymbol{I}$. The strength of all interactions was normalized by $\boldsymbol{I} \leftarrow \boldsymbol{I}/\max_i \boldsymbol{I}_i$ to compute their relative strength. Then, we drew the curve of the relative strength of interactions by sorting them in descending order. Figure 3 shows the average curve over different input samples. **We found that using the principal feature components could significantly enhance the sparsity of interactions.** Explanations based on such sparser interactions were more likely to represent the primitives for the inference logic of a DNN.

### 3.2.2 UNIVERSAL APPROXIMATION PROPERTY

In this section, we aim to verify that interactions defined on feature components can accurately mimic the entire model output $g(f)$. We also followed the settings in the "*Experimental setting*" paragraph to extract AND-OR interactions. Let $\Omega_\alpha$ denote the set of the most salient $\alpha$ interactions with the largest values of $|I_{\text{and/or}}(S|f)|$. Then, we used the metric $\mathbb{E}_x|\Delta g(f)| = \mathbb{E}_f[|g(f) - \hat{g}_\alpha(f)|]$, *w.r.t.* $\hat{g}_\alpha(f) = g(f_\emptyset) + \sum_{S \in \Omega_\alpha} I_{\text{and}}(S|f) + \sum_{S \in \Omega_\alpha} I_{\text{or}}(S|f)$, to measure the approximation error of using different numbers $\alpha$ of interactions. Given each input, we computed the least number of interactions $\hat{\alpha}$ that were required to cover 90% of the network output $g(f)$, *i.e.,* $\hat{\alpha} = \min \alpha$ s.t. $(|g(f) - \hat{g}_\alpha(f)|)/|g(f)| \le 0.1$. Figure 4 reports the average approximation error over different samples and the average ratio of the minimum interaction number ($E_f[\hat{\alpha}/2^{r+1}]$), which shows that the network outputs were usually well approximated by only using less than 10% most salient interactions.

We also conducted experiments to demonstrate that the sum of a few interactions could well approximate various network outputs on an exponential number of randomly masked features $\{g(f_S)\}_S$. Specifically, we measured the approximation error when we used the most salient $\alpha \in \{20, 40, 60\}$ interactions, respectively. Then, we computed $\Delta g_\alpha(f_S) = g(f_S) - \hat{g}_\alpha(f_S)$ as the approximation error on $f_S$, where $\hat{g}_\alpha(f_S) = g(f_\emptyset) + \sum_{L \in \Omega_\alpha, \emptyset \ne L \subseteq S} I_{\text{and}}(L|f) + \sum_{L \in \Omega_\alpha, L \cap S \ne \emptyset} I_{\text{or}}(L|f)$. Figure 5 shows network outputs on all $2^n$ masked features in an ascending order, and the shaded area represents the approximation error. For visualization, we averaged the approximation error over 50 neighboring masked features for smoothing. The results show that a small number (usually less than 60) of interactions could well approximate the varying network outputs on different masked features.

## 4 VISUALIZATION OF THE EMERGENCE OF PRIMITIVE INTERACTIONS

### 4.1 VISUALIZATION OF INTERACTIONS ON FEATURES

**Visualization of principal feature components in the interaction.** As the first step of visualizing the interaction, we visualize each principal feature component $f_i$ involved in the interaction. We draw a heatmap on raw input variables of the sample $x$ that corresponds to each feature component $f_i$. Different types of data require different visualization methods, which are as follows.

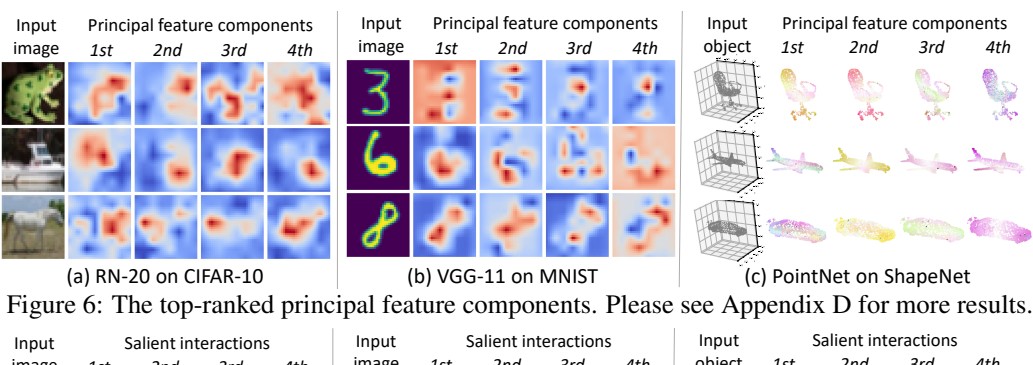

Figure 6: The top-ranked principal feature components. Please see Appendix D for more results.

Figure 7: Visualization of salient interactions. Please see Appendix D for more results.

*For image data.* For an input image, let $f \in \mathbb{R}^{H \times W \times C}$ denote its intermediate-layer feature, where $H, W$, and $C$ denotes the height, width, and channel number of the feature map, respectively. Let $q_i \in \mathbb{R}^{H \times W \times C}$ denote the $i$-th principal feature direction of the feature tensor. We consider $f$ to be a map with $H \times W$ positions, and the feature at each position is denoted by $p_j \in \mathbb{R}^C (j \in \{1, 2, \ldots, H \times W\})$. Correspondingly, each feature direction vector $q_i$ can also be divided into $H \times W$ vectors for $H \times W$ positions, *i.e.*, $q_{i,j} \in \mathbb{R}^C (j \in \{1, 2, \ldots, H \times W\})$. Then, we compute $q_{i,j}^T(p_j - \bar{p}_j) \in \mathbb{R}$ to represent the significance of the feature component along the $i$-th feature direction at the $j$-th position, where $\bar{p}_j$ is computed by averaging the positional features $p_j$ extracted after different epochs. In addition, $s_i = q_i^T(f - \bar{f})$ represents the overall influence of the feature $f$ along the $i$-th principal direction $q_i$. Thus, we use the metric $\mathcal{M}_j^{(i)} = \text{sign}(s_i) \cdot q_{i,j}^T(p_j - \bar{p}_j)$ to represent the influence of the $j$-th position in the feature map on strengthening the overall influence $|s_i|$ of the $i$-th feature component.

In this way, we obtain a heatmap $\mathcal{M}^{(i)} \in \mathbb{R}^{H \times W}$ for each feature component, and we further rescale the heatmap to the input size for visualization. Figure 6 (a,b) shows the resulting heatmaps, where the highlighted regions represent patches that significantly influence the $i$-th feature component $f_i$.

*For point cloud data.* Just like (Simonyan et al., 2014), we visualize the gradient of the significance of feature components *w.r.t.* 3D points in the input. Given a point cloud $x \in \mathbb{R}^{n \times 3}$, let $x_j \in \mathbb{R}^3$ denote a 3D point, and let $f \in \mathbb{R}^m$ denote the intermediate-layer feature. For each feature direction $q_i$, we use $|s_i|$ to represent the significance of the $i$-th feature component in $f$, *i.e.*, $s_i = q_i^T(f - \bar{f})$. Then, we compute the gradient of $|s_i|$ *w.r.t.* the 3D coordinates of each 3D point, $\mathcal{M}_j^{(i)} = \partial |s_i| / \partial x_j \in \mathbb{R}^3$ to represent the influence of the $j$-th 3D point on the $i$-th feature component. To visualize the influence of different 3D points on the $i$-th feature component, we use the RGB color channels to represent the three-dimensional gradient $\mathcal{M}^{(i)}$. We also normalize the gradients over different 3D points to the range $[0, 1]$ for visualization (please refer to the Appendix D for more details). Figure 6 (c) shows the heatmaps of the top-4 feature components in PointNet trained on the ShapeNet dataset.

*For language data.* We use the Shapley value (Shapley, 1953) to measure the attribution of each word in the input sentence to the principal feature component. Given the feature $f$ of the input sentence $x$, we compute $s_i = q_i^T(f - \bar{f})$ to represent the influence of the $i$-th feature component. Then, we compute the following Shapley value to measure the contribution of each word in the input to the influence $s_i$ of the $i$-th feature component. Let $N_{\text{word}} = \{1, 2, \ldots, n\}$ denote indices of all words in the input sentence. The Shapley value $\mathcal{M}_j^{(i)}$ of each word $x_j (j \in N_{\text{word}})$ measures the numerical contribution of the word $x_j$ to the influence $s_i$ of the $i$-th feature component, as follows.

$$\mathcal{M}_j^{(i)} = \sum_{T \subseteq N_{\text{word}} \setminus \{j\}} \left[ |T|!(n - |T| - 1)!/n! \right] \cdot \left[ s_i(x_{T \cup \{j\}}) - s_i(x_T) \right] \quad (5)$$

where $s_i(x_T) = q_i^T(f_{|x_T} - \bar{f})$ denotes the influence of the $i$-th principal feature component when we input a masked sentence $x_T$ to the DNN. $f_{|x_T}$ denotes the intermediate-layer feature extracted from

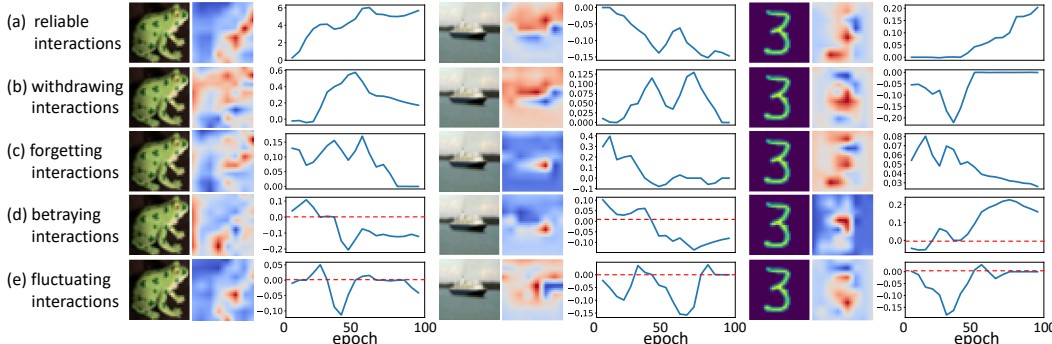

Figure 8: Visualization of three salient interactions in training samples of the SST-2 dataset, and how they are learned during the training process.

Figure 9: Curves of the utility of interactions during the learning of DNNs. These interactions can be categorized into five groups. Please refer to Appendix E for results on more samples.

the masked sentence $x_T$. In this way, the Shapley values of different words serve as an attribution map of different words to the $i$-th principal feature component $f_i$. Please refer to Appendix D for the visualization on the language data.

**Visualization of interactions.** Then, we visualize the interaction, which is composed of multiple feature components. For each interaction $S \subseteq N_{\text{feature}}$ with a considerable utility $I_{\text{and/or}}(S|x)$, we accumulate the heatmaps $\{\mathcal{M}^{(i)} | i \in S\}$ of all the principal feature components involved in $S$ to visualize the interaction. Specifically, for each principal feature component $i \in S$, we first normalize its heatmap $\mathcal{M}^{(i)}$ by $\mathcal{M}'^{(i)} = \mathcal{M}^{(i)} / \max_j(|\mathcal{M}_j^{(i)}|)$. Then, we sum up the heatmaps of all the feature components involved in interaction $S$ to visualize interaction $S$, *i.e.*, $\mathcal{M}^{(S)} = \sum_{i \in S} \mathcal{M}'^{(i)}$.

Figure 7 visualizes interactions with the largest utility $|I_{\text{and/or}}(S|x)|$ in RN-20 and PointNet trained on the MNIST, CIFAR-10, and ShapeNet datasets. Figure 8 (b) visualizes interactions on the SST-2 dataset. The red color indicates that the word has a positive attribution to the principal feature component, while the blue color indicates a negative attribution.

## 4.2 EMERGENCE OF PRIMITIVE INTERACTIONS DURING THE TRAINING PROCESS

In this section, we visualize the dynamics of interactions during the learning process. Then, we find that the salient interactions encoded in the DNN usually can be categorized into five groups, *i.e., reliable interactions, withdrawing interactions, forgetting interactions, betraying interactions, and fluctuating interactions*. The number and complexity of interactions in each group provide new insights into the learning of the DNN.

Specifically, let $\theta_0, \theta_1, \ldots, \theta_T$ denote parameters of DNNs trained after different epochs $t = 0, 1, \ldots, T$. We use $I_{\text{and/or}}(S|x, \theta_t)$ to denote the interaction in the DNN after $t$ epochs. In order to visualize the dynamics of the interactions during training, we draw the curve of $I_{\text{and/or}}(S|x)$ for each salient interaction $S$ across different epochs. For each DNN, we compute the interaction utility based on the top $r = 10$ principal feature components after every five epochs. Figure 9 illustrates the curves of $I_{\text{and/or}}(S|x)$ for each salient interaction throughout the learning process.

**Five groups of interactions.** Figure 9 (a,b) shows the interactions belonging to the first and second groups, respectively. In the first group, the strength of the utility of these interactions increases throughout the learning process in a relatively stable manner. Thus, we can consider such interac-

Figure 10: The number of interactions of each order in each group.

Table 1: Average number of interactions belonging to each group in the most salient interactions.

|  | Reliable | Withdrawing | Forgetting | Betraying | Fluctuating |
|---|---|---|---|---|---|
| VGG-11 on CIFAR-10 | 28.4 | **26.4** | 6.0 | **26.6** | **12.6** |
| RN-20 on CIFAR-10 | **33.4** | 26.2 | **16.6** | 17.8 | 6.0 |
| VGG-11 on MNIST | 44.2 | **21.4** | 5.8 | **20.8** | **7.8** |
| RN-20 on MNIST | **49.6** | 18.2 | **18.0** | 12.6 | 1.6 |
| CNN on SST-2 | 33.6 | 14.4 | 0.4 | 12.8 | 38.8 |

tions are stably learned by the DNN, and we call them *reliable interactions*. In the second group, the utility of interactions is usually close to zero in the beginning. Then, the strength of their utility first increases and then decreases, sometimes decreasing to almost zero. These interactions are referred to as *withdrawing interactions*.

As Figure 9 (c) shows, the initial utility of interactions in the third group is non-ignorable. In the third group, the interaction strength keeps decreasing to zero. These interactions are gradually forgotten by the DNN. We call them *forgetting interactions*.

Figure 9 (d,e) show interactions in the fourth and fifth groups. In the fourth group, interactions experience a gradual shift towards an interaction utility that is opposite to their initial utility. These interactions are called *betraying interactions*. The interactions in the fifth group have fluctuating interactive utilities throughout the learning process, being called *fluctuating interactions*.

**The number and order (complexity) of interactions in each group.** For each DNN, we selected 100 interactions whose maximum interactions strength ($\max_t |I_{and}(S|x,\theta_t)|$ and $\max_t |I_{or}(S|x,\theta_t)|$) throughout the training process were ranked in top 100 among all interactions. Then, we computed the number of interactions belonging to each group among these 100 salient interactions. Table 1 reports the average number of interactions in each group over different samples. We found that compared to VGG-11, RN-20 learned more reliable and forgetting interactions, while having less betraying and fluctuating interactions. This might be because the residual connections in ResNet-20 made the features more stable. Besides, we also noticed that the DNNs trained on the MNIST dataset usually encoded more reliable interactions and less betraying and fluctuating interactions than DNNs trained on the CIFAR-10 dataset. This result indicated that the dynamics of interactions provided us with a new perspective to analyze the difficulty of training a DNN on a dataset.

We further studied the complexity (order) of interactions in each group. Figure 10 reports the number of interactions of each order in different groups. We found that the distribution of interactions over different orders were similar in different models. Besides, we found that high-order interactions were usually fluctuating and withdrawing interactions, because high-order interactions usually represented complex unstable features.

## 5 CONCLUSION

In this study, we have proposed a method for visualizing the changes in interactions encoded in a DNN during the learning process. We have extended the interaction defined on raw input variables by (Li & Zhang, 2023a), and have newly defined interactions on principal feature components. This extension greatly boosts the sparsity/simplicity of the interaction-based explanation of a DNN, which enables us to visualize how different interactions gradually emerge during the learning process. Based on the visualization of interactions throughout the entire learning process, we have found that all interactions could be categorized into five groups, *i.e.,* reliable, withdrawing, forgetting, betraying, and fluctuating interaction. The visualization of how a DNN learns different types of interactions offers a novel perspective for understanding the DNN.

ETHIC STATEMENT

In this study, we propose a method to define and visualize the emergence of primitive interactions throughout the learning of a DNN, which can help us understand how a DNN learns features from the dataset during the training process. The proposed interactions can be considered as primitive inference patterns encoded by the DNN, and can faithfully explain the output scores of the DNN. There are no ethical issues with this paper.

REPRODUCIBILITY STATEMENT

We have provided proofs for the theoretical results of this study in Appendix B and Appendix C. We have also provided experimental details in the *"Experimental settings"* paragraph in Section 3.2 and Appendix D.

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

## A  COMMON CONDITIONS FOR SPARSE INTERACTIONS

Ren et al. (2023c) have proven that under three common conditions, a trained DNN usually encodes very sparse interactions. The three conditions are as follows. (1) The high-order derivatives of the network output *w.r.t.* the input are all zero; (2) When we randomly mask some input variables, the average confidence of the network inference over different masked samples monotonically decreases along with the number of masked input variables. (3) The decreasing speed of the average inference confidence is not faster than a polynomial function along with the ratio of masked input variables.

## B  RELATIONSHIP BETWEEN AND INTERACTIONS AND OR INTERACTIONS

In this section, we provide a further discussion on understanding the relationship between AND and OR interactions.

Given a DNN $v(\cdot)$ and an input sample $x \in \mathbb{R}^n$, let $b \in \mathbb{R}^n$ denote the baseline values input variables, which represent the masked states of variables in the input. Then, if the variables in $L \subseteq N$ are preserved and other variables are masked, the masked input $x_L$ is defined as follows.

$$(x_L)_i = \begin{cases} x_i, & i \in L \\ b_i, & i \notin L \end{cases} \tag{6}$$

Based on the above definition of masked inputs, the AND interaction can be computed as $I_{\text{and}}(S|x) = \sum_{L \subseteq S}(-1)^{|S|-|L|}v_{\text{and}}(x_L)$. The OR interaction between variables in $x$ is computed as $I_{\text{or}}(S|x) = -\sum_{L \subseteq S}(-1)^{|S|-|L|}v_{\text{or}}(x_{N \setminus L})$. To simplify the analysis, we assume $v_{\text{and}}(\cdot) = v_{\text{or}}(\cdot) = 0.5v(\cdot)$.

Conversely, if we consider $b$ as the input sample and take $x$ as baseline values of input variables in $b$, the masked input $b_L$ is defined as follows.

$$(b_L)_i = \begin{cases} b_i, & i \in L \\ x_i, & i \notin L \end{cases} \tag{7}$$

According to Eq. (6) and Eq. (7), we have $x_{N \setminus L} = b_L$. Therefore, we can rewrite the OR interaction as follows.

$$\begin{aligned} I_{\text{or}}(S|x) &= -\sum_{L \subseteq S}(-1)^{|S|-|L|}v_{\text{or}}(x_{N \setminus L}) \\ &= -\sum_{L \subseteq S}(-1)^{|S|-|L|}v_{\text{or}}(b_L) \\ &= -\sum_{L \subseteq S}(-1)^{|S|-|L|}v_{\text{and}}(b_L) \\ &= -I_{\text{and}}(S|b) \end{aligned} \tag{8}$$

Therefore, the OR interaction can be viewed as a special AND interaction by considering original variable values $x$ as masked states and taking the masked states $b$ as normal values of the variables.

## C  EXPLAINING ATTRIBUTIONS OF PRINCIPAL COMPONENTS

In this section, we prove that we can directly use the AND and OR interactions to estimate Shapley values (Shapley, 1953) of principal feature components, which also demonstrates the reliability of the extracted interactions.

The Shapley value is one of the most classic metrics for measuring the numerical contribution (or attribution) of each input variable, *i.e.,* the attribution of each principal feature component to the DNN output $g(f)$. In the scenario of Shapely values, we consider the $r$ feature components $f_1, f_2, \ldots, f_r(f_i = q_i q_i^T(f - \bar{f}))$ as $r$ players. The inference score $g(f)$ is considered as the overall reward obtained by all players (feature components). The Shapley value is proposed to allocate the overall reward $g(f)$ to each player (feature component). Thus, we can consider the Shapley value

of the feature component $f_i$ as the attribution of $f_i$ to the DNN output. The Shapley value of the feature component $f_i$ is given as follows.

$$\phi(i|f) = \sum_{S \subseteq N_{\text{feature}} \setminus \{i\}} \left[ |S|!(r - |S| - 1)!/r! \right] \cdot \left[ g(f_{S \cup \{i\}}) - g(f_S) \right] \quad (9)$$

The Shapley value has been proven (Weber, 1988) to satisfy the *linearity, nullity, symmetry*, and *efficiency* axioms, thus being regarded as a fair metric for allocating the reward (*i.e.*, the network output $g(f)$) to different feature components.

**Theorem 2** (proven in the supplementary material). *The Shapley value $\phi(i|f)$ of each feature component $f_i$ can be reformulated as $\phi(i|f) = \sum_{S \subseteq N_{\text{feature}}, i \in S} \frac{1}{|S|} I_{and}(S|f) + \sum_{S \subseteq N_{\text{feature}}, i \in S} \frac{1}{|S|} I_{or}(S|f)$.*

### C.1    PROOF OF THEOREM 2

*Proof:* According to the definition of Shapley values, $\phi(i|f) = \mathbb{E}_{S \subseteq N_{\text{feature}} \setminus \{i\}} [g(f_{S \cup \{i\}}) - g(f_S)]$. For simplicity, we use $g(S)$ to represent the output $g(f_S)$ on the masked feature $f_S$. Similarly, we use $I_{\text{and}}(S)$ and $I_{\text{or}}(S)$ to represent $I_{\text{and}}(S|f)$ and $I_{\text{or}}(S|f)$, respectively. Besides, we use $N$ to represent $N_{\text{feature}}$.

Then, according to Theorem 1 in the paper, we have $\forall S \subseteq N, g(S) = g(\emptyset) + \sum_{L \subseteq S, L \neq \emptyset} I_{\text{and}}(L) + \sum_{L \cap S \neq \emptyset} I_{\text{or}}(L)$. Thus,

$$g(S \cup \{i\}) - g(S)$$

$$= \left[ g(\emptyset) + \sum_{L \subseteq (S \cup \{i\}), L \neq \emptyset} I_{\text{and}}(L) + \sum_{L \cap (S \cup \{i\}) \neq \emptyset} I_{\text{or}}(L) \right] - \left[ g(\emptyset) + \sum_{L \subseteq S, L \neq \emptyset} I_{\text{and}}(L) + \sum_{L \cap S \neq \emptyset} I_{\text{or}}(L) \right]$$

$$= \left[ \sum_{L \subseteq (S \cup \{i\}), L \neq \emptyset} I_{\text{and}}(L) - \sum_{L \subseteq S, L \neq \emptyset} I_{\text{and}}(L) \right] + \left[ \sum_{L \cap (S \cup \{i\}) \neq \emptyset} I_{\text{or}}(L) - \sum_{L \cap S \neq \emptyset} I_{\text{or}}(L) \right]$$

$$= \underbrace{\sum_{L \subseteq S} I_{\text{and}}(L \cup \{i\})}_{\mathcal{A}} + \underbrace{\sum_{L \cap S = \emptyset} I_{\text{or}}(L \cup \{i\})}_{\mathcal{B}}$$

In this way, the Shapley value can be decomposed into $\phi(i|f) = \mathbb{E}_{S \subseteq N \setminus \{i\}}[\mathcal{A} + \mathcal{B}]$. In the following proof, we first analyze the sum of AND interactions $\mathbb{E}_{S \subseteq N \setminus \{i\}}[\mathcal{A}]$, and then analyze the sum of OR interactions $\mathbb{E}_{S \subseteq N \setminus \{i\}}[\mathcal{B}]$.

$$\mathbb{E}_{S \subseteq N \setminus \{i\}}[\mathcal{A}]$$

$$= \mathbb{E}_{S \subseteq N \setminus \{i\}} \sum_{L \subseteq S} I_{\text{and}}(L \cup \{i\})$$

$$= \frac{1}{n} \sum_{m=0}^{n-1} \frac{1}{\binom{n-1}{m}} \sum_{\substack{S \subseteq N \setminus \{i\}, \ L \subseteq S \\ |S| = m}} I_{\text{and}}(L \cup \{i\})$$

$$= \frac{1}{n} \sum_{L \subseteq N \setminus \{i\}} \sum_{m=0}^{n-1} \frac{1}{\binom{n-1}{m}} \sum_{\substack{S \supseteq L, \\ S \subseteq N \setminus \{i\}, \\ |S| = m}} I_{\text{and}}(L \cup \{i\})$$

$$= \frac{1}{n} \sum_{L \subseteq N \setminus \{i\}} \sum_{m=|L|}^{n-1} \frac{1}{\binom{n-1}{m}} \sum_{\substack{S \supseteq L, \\ S \subseteq N \setminus \{i\}, \\ |S| = m}} I_{\text{and}}(L \cup \{i\}) \quad \text{// since } S \supseteq L, |S| = m \geq |L|.$$

$$= \frac{1}{n} \sum_{L \subseteq N \setminus \{i\}} \sum_{m=|L|}^{n-1} \frac{1}{\binom{n-1}{m}} \binom{n-1-|L|}{m-|L|} I_{\text{and}}(L \cup \{i\})$$

$$=\frac{1}{n}\sum_{L\subseteq N\setminus\{i\}}\underbrace{\sum_{k=0}^{n-1-|L|}\frac{1}{\binom{n-1}{|L|+k}}\binom{n-1-|L|}{k}}_{\alpha_L}I_{\text{and}}(L\cup\{i\})\quad\text{// Let }k=m-|L|.$$

$$=\sum_{L\subseteq N\setminus\{i\}}\frac{1}{|L|+1}I_{\text{and}}(L\cup\{i\})\quad\text{// Ren et al. (2023a) have proven that }\alpha_L=\frac{n}{|L|+1}.$$

$$=\sum_{S\subseteq N,i\in S}\frac{1}{|S|}I_{\text{and}}(S)\quad\text{// Let }S=L\cup\{i\}.$$

Then, for the sum of OR interactions, we have

$$\mathbb{E}_{S\subseteq N\setminus\{i\}}[\mathcal{B}]$$

$$=\mathbb{E}_{S\subseteq N\setminus\{i\}}\sum_{L\cap S\neq\emptyset}I_{\text{or}}(L\cup\{i\})$$

$$=\frac{1}{n}\sum_{m=0}^{n-1}\frac{1}{\binom{n-1}{m}}\sum_{\substack{S\subseteq N\setminus\{i\},\,L\cap S\neq\emptyset\\|S|=m}}I_{\text{or}}(L\cup\{i\})$$

$$=\frac{1}{n}\sum_{L\subseteq N\setminus\{i\}}\sum_{m=0}^{n-1}\frac{1}{\binom{n-1}{m}}\sum_{\substack{S\cap L\neq\emptyset,\\S\subseteq N\setminus\{i\},\\|S|=m}}I_{\text{or}}(L\cup\{i\})$$

$$=\frac{1}{n}\sum_{L\subseteq N\setminus\{i\}}\sum_{m=0}^{n-1}\frac{1}{\binom{n-1}{m}}\sum_{\substack{S\subseteq N\setminus\{i\}\setminus L,\\|S|=m}}I_{\text{or}}(L\cup\{i\})$$

$$=\frac{1}{n}\sum_{L\subseteq N\setminus\{i\}}\sum_{m=0}^{n-1-|L|}\frac{1}{\binom{n-1}{m}}\sum_{\substack{S\subseteq N\setminus\{i\}\setminus L,\\|S|=m}}I_{\text{or}}(L\cup\{i\})\quad\text{// Since }S\subseteq N\setminus\{i\}\setminus L,|S|\leq n-1-|L|.$$

$$=\frac{1}{n}\sum_{L\subseteq N\setminus\{i\}}\sum_{m=0}^{n-1-|L|}\frac{1}{\binom{n-1}{m}}\binom{n-1-|L|}{m}I_{\text{or}}(L\cup\{i\})$$

$$=\frac{1}{n}\sum_{L\subseteq N\setminus\{i\}}\sum_{k=0}^{n-1-|L|}\frac{1}{\binom{n-1}{n-1-|L|-k}}\binom{n-1-|L|}{n-1-|L|-k}I_{\text{or}}(L\cup\{i\})\quad\text{// Let }k=n-1-|L|-m.$$

$$=\frac{1}{n}\sum_{L\subseteq N\setminus\{i\}}\underbrace{\sum_{k=0}^{n-1-|L|}\frac{1}{\binom{n-1}{|L|+k}}\binom{n-1-|L|}{k}}_{\alpha_L}I_{\text{or}}(L\cup\{i\})$$

$$=\frac{1}{n}\sum_{L\subseteq N\setminus\{i\}}\frac{n}{|L|+1}I_{\text{or}}(L\cup\{i\})$$

$$=\sum_{L\subseteq N\setminus\{i\}}\frac{1}{|L|+1}I_{\text{or}}(L\cup\{i\})$$

$$=\sum_{S\subseteq N,i\in S}\frac{1}{|S|}I_{\text{or}}(S)\quad\text{// Let }S=L\cup\{i\}.$$

Therefore, $\phi(i|f)=\sum_{S\subseteq N\setminus\{i\}}[\mathcal{A}]+\sum_{S\subseteq N\setminus\{i\}}[\mathcal{B}]=\sum_{S\subseteq N,i\in S}\frac{1}{|S|}I_{\text{and}}(S)+\sum_{S\subseteq N,i\in S}\frac{1}{|S|}I_{\text{or}}(S)$.

## C.2 EXPERIMENTAL VERIFICATION

Additionally, we conducted experiments to examine the above mathematical connection, *i.e.*, using the utility of all AND interactions and OR interactions to estimate Shapley values of principal feature components. Let $\Omega_\alpha$ denote the most salient $\alpha$ interactions with the largest $|I_{\text{and/or}}(S|f)|$. Then, we used all AND/OR interactions in $\Omega_\alpha$ to estimate the Shapley value of the $i$-th feature component, as follows. $\hat{\phi}_\alpha(i|f) = \sum_{S \subseteq N_{\text{feature}}, i \in S, S \in \Omega_\alpha} \frac{1}{|S|} I_{\text{and}}(S|f) + \sum_{S \subseteq N_{\text{feature}}, i \in S, S \in \Omega_\alpha} \frac{1}{|S|} I_{\text{or}}(S|f)$. Figure 11 shows the cosine similarity between the accurate Shapley values $\phi = [\phi(1|f), \phi(2|f), \ldots, \phi(r|f)]$ and the Shapley values $\hat{\phi} = [\hat{\phi}(1|f), \hat{\phi}(2|f), \ldots, \hat{\phi}(r|f)]$ estimated using the top $\alpha$ salient interactions. The results in Figure 11 indicate that the top-ranked 10% interactions had provided sufficient information to well explain the Shapley values.

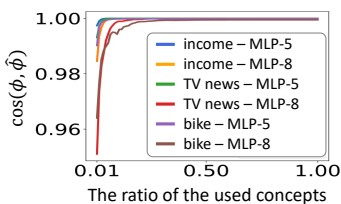

Figure 11: The cosine similarity between the accurate Shapley values and the Shapley values estimated using interactions.

## D MORE VISUALIZATION RESULTS OF PRINCIPAL FEATURE COMPONENTS AND SALIENT INTERACTIONS

This section provides additional results of the visualization of principal feature components and salient interactions.

For image data, Figures 12 and 13 show the visualization of principal feature components extracted from the ResNet-20 and VGG-11 models trained on the CIFAR-10 dataset, respectively. Using these feature components, Figures 14 and 15 visualize the salient interactions encoded in the two models. Similarly, Figures 16 and 17 show the visualization of principal feature components obtained from the ResNet-20 and VGG-11 trained on the MNIST dataset, respectively, and Figures 18 and 19 visualize the salient interactions encoded in these models.

For language data, we trained a three-layer unidirectional LSTM and a CNN on the SST-2 dataset, respectively. Figure 20 shows both visualizations of feature components and salient interactions in the LSTM network. Figure 21 shows both visualizations of feature components and salient interactions in the CNN network. We found that most salient interactions in language data usually only consisted of only a few top-ranked feature components. As a result, the visualization of salient interactions was sometimes similar to the visualization of feature components.

For point cloud data, we visualize the gradient of the significance of feature components *w.r.t.* 3D points in the input. Given a point cloud $x \in \mathbb{R}^{n \times 3}$, let $x_j \in \mathbb{R}^3$ denote a 3D point, and let $f \in \mathbb{R}^m$ denote the intermediate-layer feature. For each feature direction $q_i$, we use $|s_i|$ to represent the significance of the feature component in $f$ along the feature direction $q_i$, where $s_i = q_i^T(f - \bar{f})$. In order to compute the gradient of $|s_i|$ *w.r.t.* the 3D coordinates of each 3D point, We first add a random noise $\delta \sim \mathcal{N}(0, 0.5^2 I) \in \mathbb{R}^{n \times 3}$ to the point cloud $x$, *i.e.,* $x^{(0)} \leftarrow x + \delta$. Then, we compute the gradient of $|s_i|$ on the perturbed input, *i.e.,* $\nabla^{(i)}(x^{(0)}) = \frac{\partial |s_i|(x^{(0)})}{\partial x^{(0)}}$. The gradient is added to the input to further strengthen the feature component by $x^{(1)} \leftarrow x^{(0)} + \eta \cdot \nabla^{(i)}(x^{(0)})$, where $\eta$ is a small constant. We iteratively compute the gradient $\nabla^{(i)}(x^{(t)})$ of the component significance *w.r.t.* the modified input $x^{(t)}$, and modify the input by $x^{(t+1)} \leftarrow x^{(t)} + \eta \cdot \nabla^{(i)}(x^{(t)})$. We use the change in the input point cloud $\mathcal{M}^{(i)} = x^{(T)} - x \in \mathbb{R}^{n \times 3}$ to represent the heatmap of the principal feature component.

In experiments, we repeatedly sampled 10 different random noises $\delta$ for each input, we computed the gradients of points in the perturbed input for $T = 20$ iterations to obtain the heatmap $\mathcal{M}^{(i)}$. To obtain a representative heatmap, we averaged the heatmaps $\mathcal{M}^{(i)}$ obtained from the 10 randomly perturbed inputs. For visualization, we took the three dimensions corresponding to each point in $\mathcal{M}^{(i)}$ as the RGB number to draw the heatmap. However, the RGB number must be in the range of $[0, 1]$. Therefore, we used two methods to normalize the values in $\mathcal{M}^{(i)}$ to the range of $[0, 1]$.

The first normalization method is to scale the value $\tilde{\mathcal{M}}_{j,c}^{(i)}$ at each point $j \in N$ in each channel $c \in \{1, 2, 3\}$ of the heatmap as $\tilde{\mathcal{M}}_{j,c}^{(i)} = |\mathcal{M}_{j,c}^{(i)}|/max_j\{\mathcal{M}_{j,c}^{(i)}\}$ for visualization. Figure 22 shows more visualization results of principal feature components in the PointNet learned on the ShapeNet dataset, and Figure 23 shows visualizations of salient interactions.

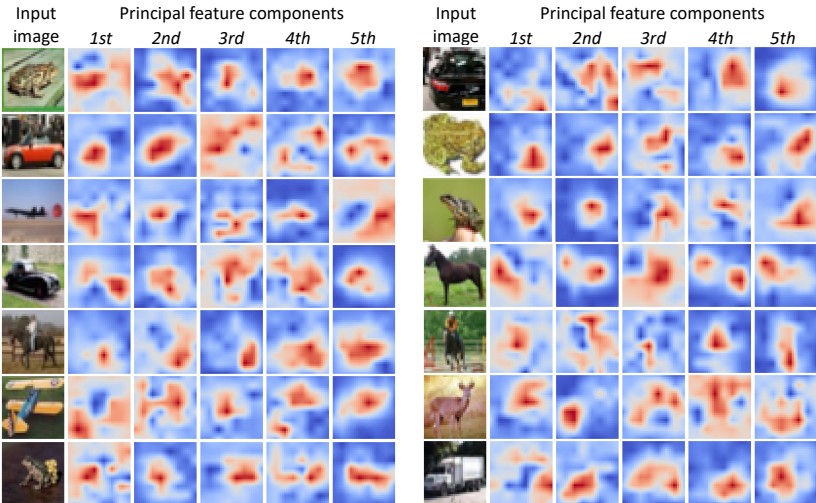

Figure 12: Visualization of principal feature components in ResNet-20 learned on the CIFAR-10 dataset.

Alternatively, the heatmap $\mathcal{M}^{(i)}$ can be normalized in another way for visualization. We computed $\tilde{\mathcal{M}}_{j,c}^{(i)} = |\mathcal{M}_{j,c}^{(i)}|/max_c\{\mathcal{M}_{j,c}^{(i)}\}$ for each point $j \in N$. Figure 24 and Figure 25 show the corresponding heatmaps of feature components and salient interactions.

# E    MORE RESULTS OF FIVE GROUPS OF INTERACTIONS

This section demonstrates more examples of interactions, which are categorized into five groups, *i.e.,* reliable, withdrawing, forgetting, betraying, and fluctuating interactions.

In order to analyze the behavior of interactions, we first computed the interaction utility $I_{\text{and/or}}(S|\theta_t)$ of each interaction $S$ at different epochs $t$. Then, all interactions were sorted based on their maximum interaction strength $\max_t |I_{\text{and/or}}(S|\theta_t)|$ throughout the learning process. We selected the top $3\%$ interactions with the largest strength $\max_t |I_{\text{and/or}}(S|\theta_t)|$ as salient interactions in the learning. These salient interactions were further categorized into five groups, as shown in Figure 27 and Figure 26. Note that not all samples exhibited all five types of interactions, because we only considered the top $3\%$ salient interactions here. Other interactions with small strengths were regarded as noisy patterns and were not discussed.

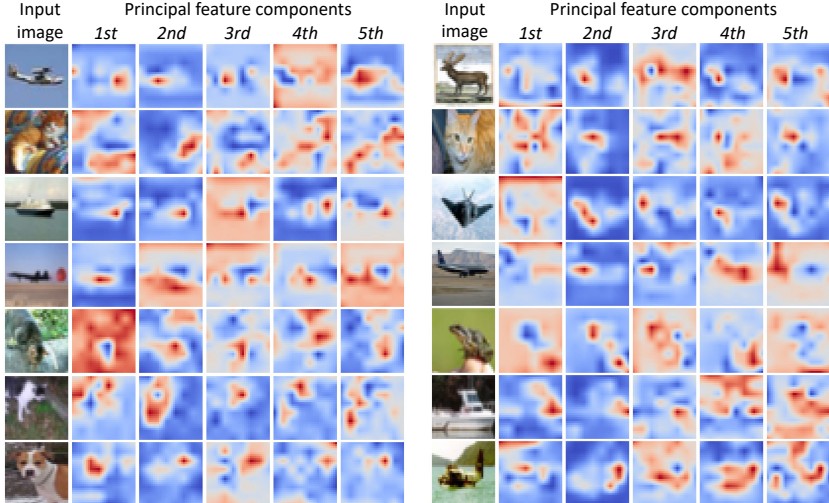

Figure 13: Visualization of principal feature components in VGG-11 learned on the CIFAR-10 dataset.

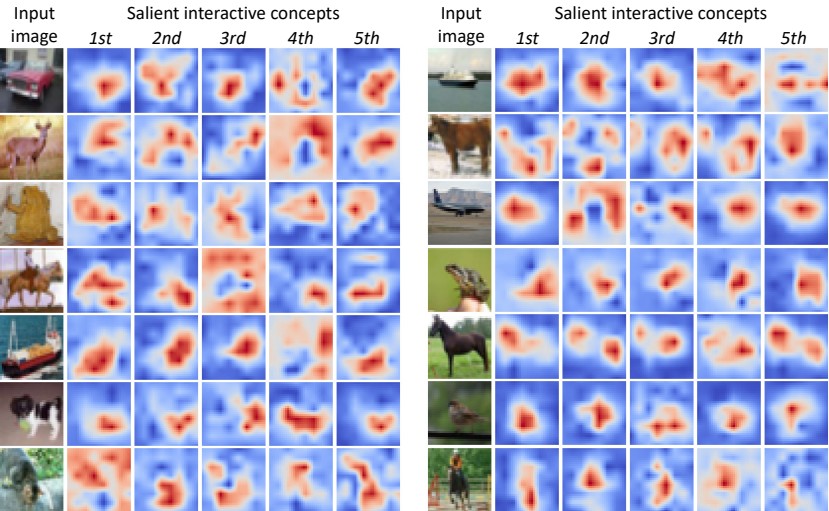

Figure 14: Visualization of salient interactions in ResNet-20 learned on the CIFAR-10 dataset.

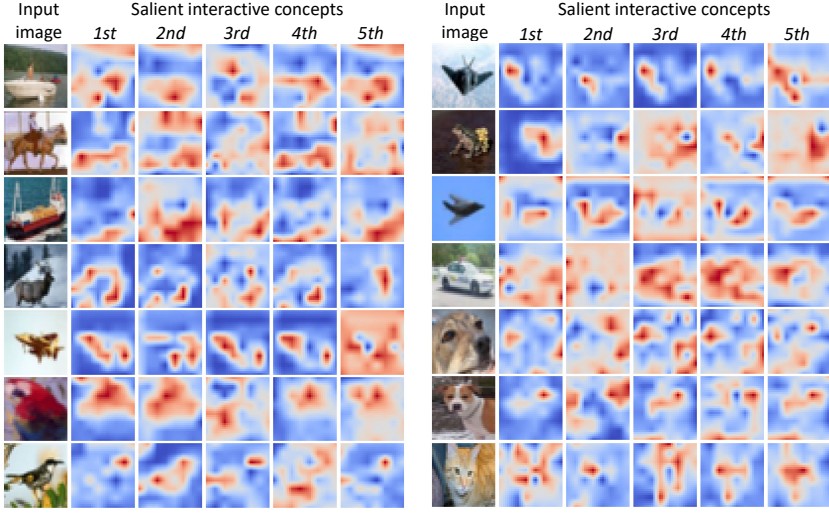

Figure 15: Visualization of salient interactions in VGG-11 learned on the CIFAR-10 dataset.

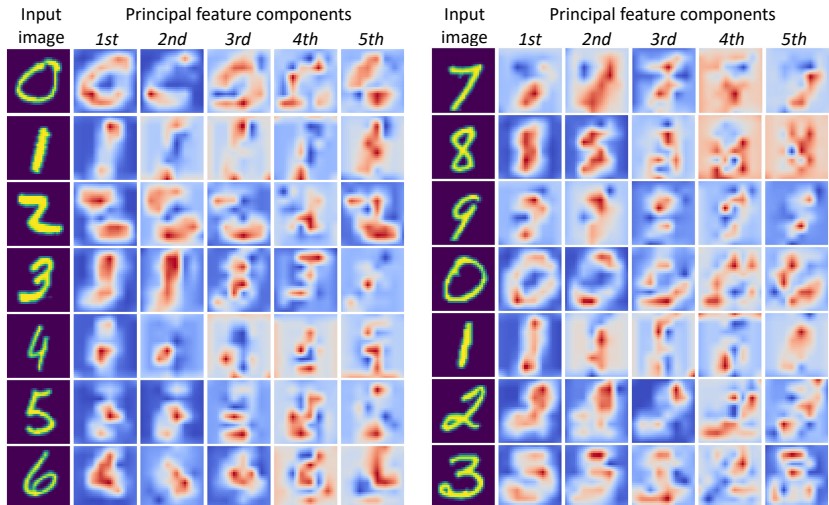

Figure 16: Visualization of principal feature components in ResNet-20 learned on the MNIST dataset.

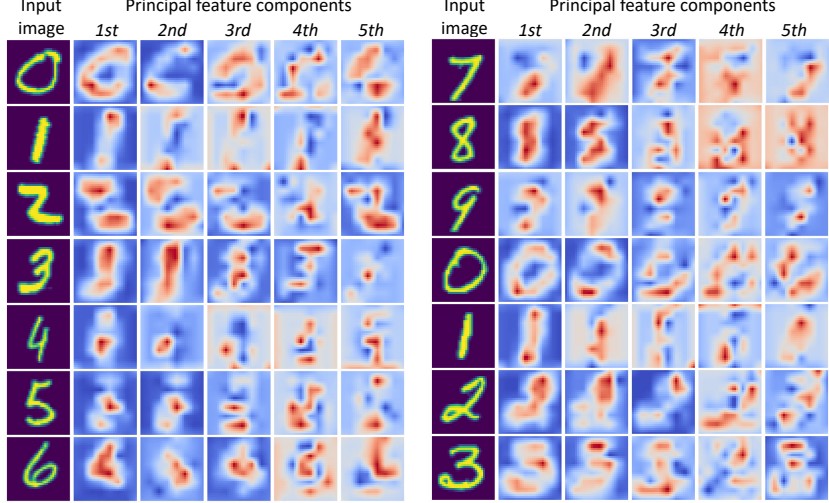

Figure 17: Visualization of principal feature components in VGG-11 learned on the MNIST dataset.

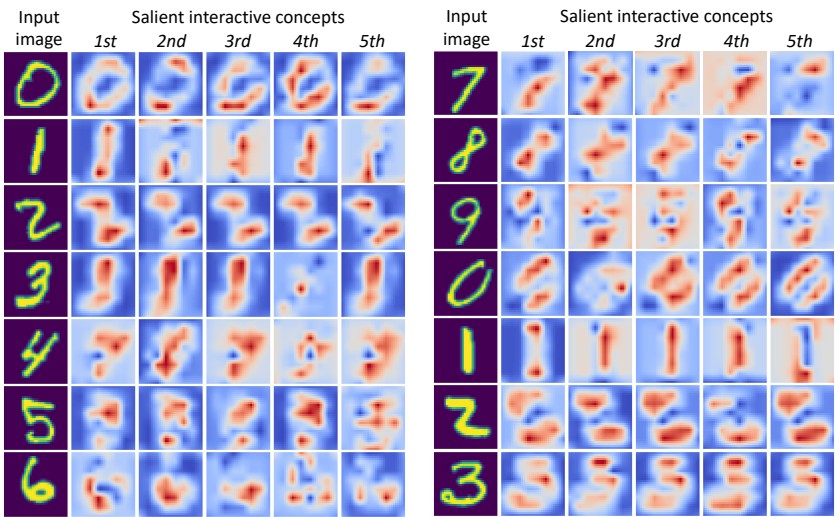

Figure 18: Visualization of salient interactions in ResNet-20 learned on the MNIST dataset.

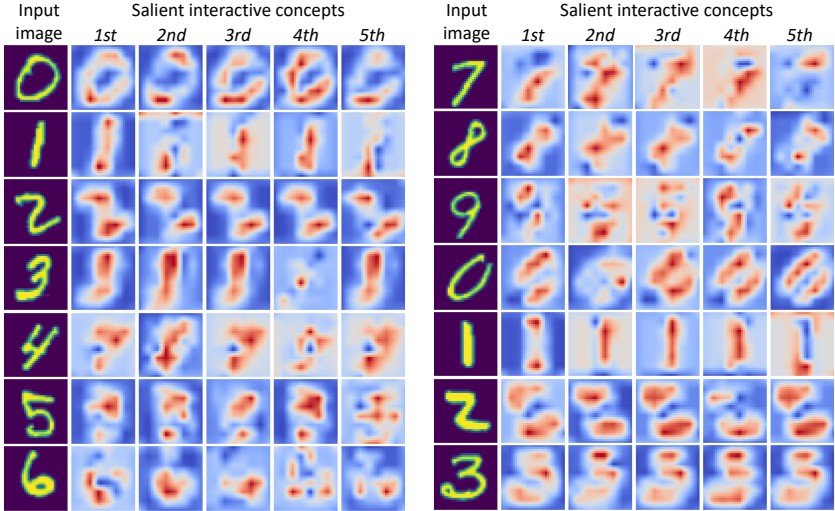

Figure 19: Visualization of salient interactions in VGG-11 learned on the MNIST dataset.

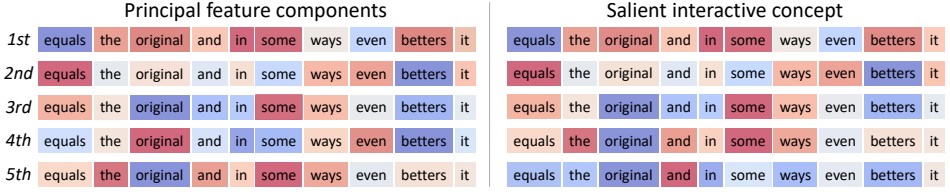

(a) The input sentence is "equals the original and in some ways even betters it", with a positive sentiment.

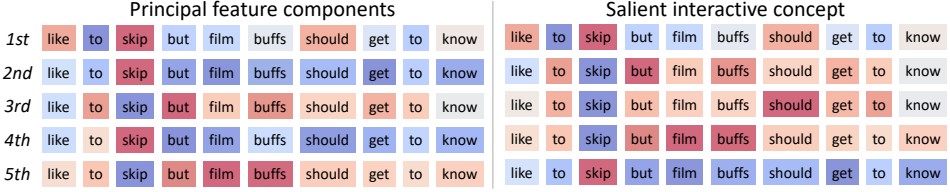

(b) The input sentence is "like to skip but film buffers should get to know", with a positive sentiment.

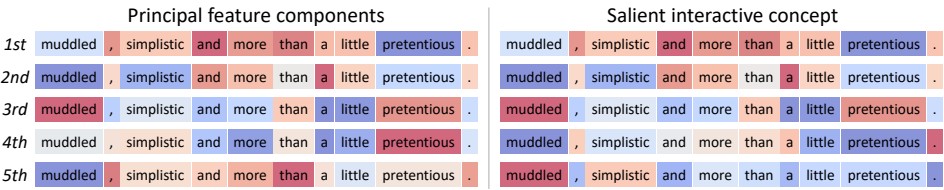

(c) The input sentence is "muddled, simplistic and more than a little pretentious.", with a negative sentiment.

Figure 20: Visualization of principal feature components and salient interactions of three examples in the LSTM trained on the SST-2 dataset.

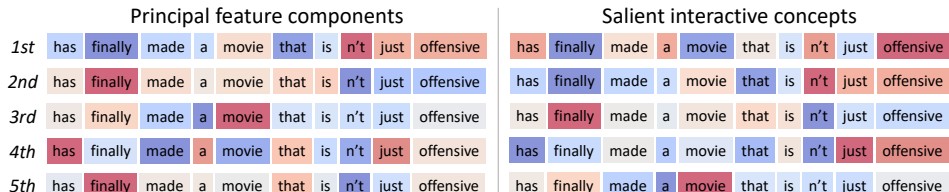

(a) The input sentence is "has finally made a movie that isn't just offensive", with a positive sentiment.

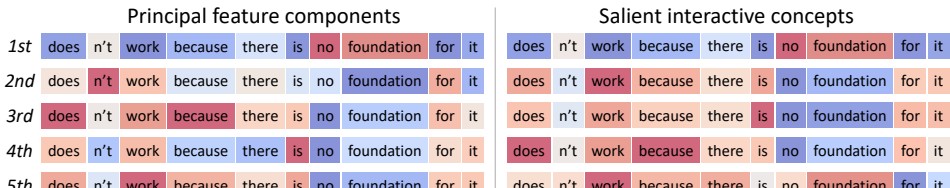

(b) The input sentence is "doesn't work because there is no foundation for it", with a negative sentiment.

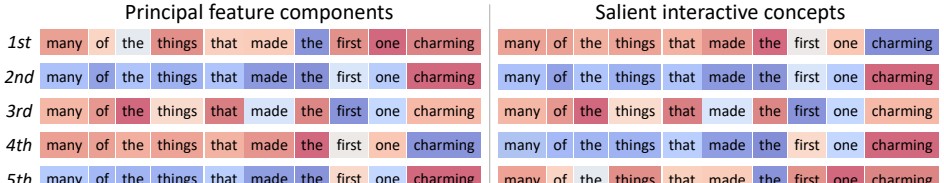

(c) The input sentence is "many of the things that made the first one charming", with a positive sentiment.

Figure 21: Visualization of principal feature components and salient interactions of three examples in the CNN trained on the SST-2 dataset.

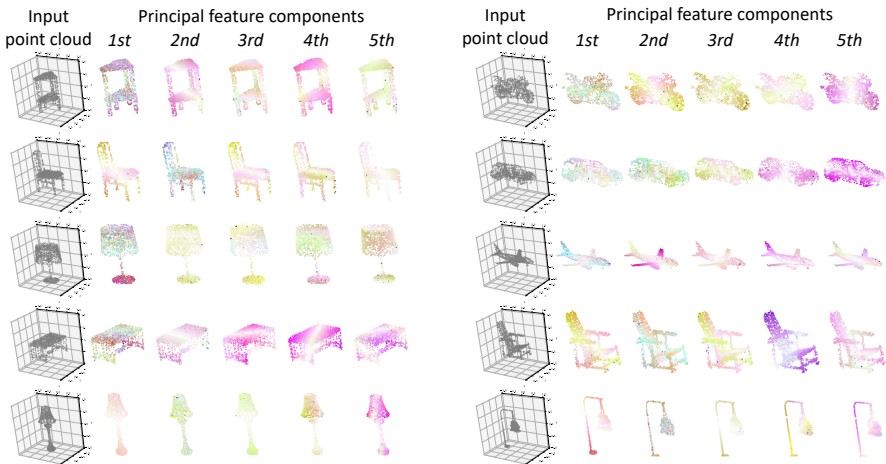

Figure 22: Visualization of principal feature components in PointNet learned on the ShapeNet dataset. Heatmaps were normalized using the first approach in Section D.

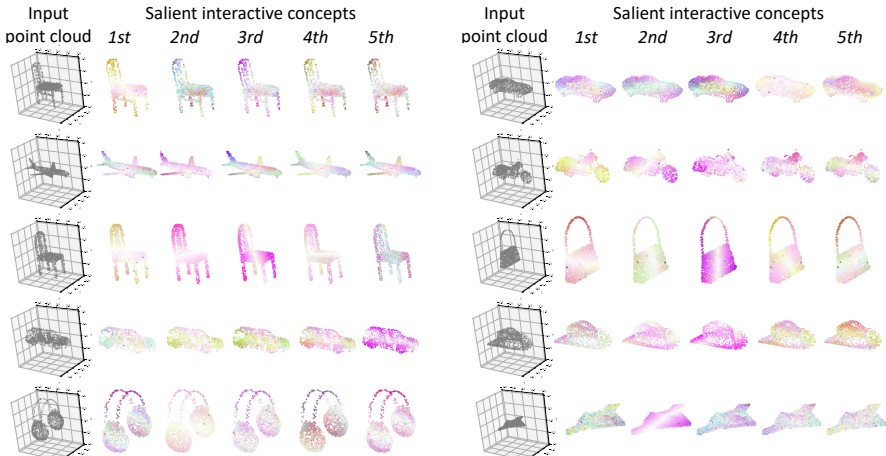

Figure 23: Visualization of salient interactions in PointNet learned on the ShapeNet dataset. Heatmaps were normalized using the first approach in Section D.

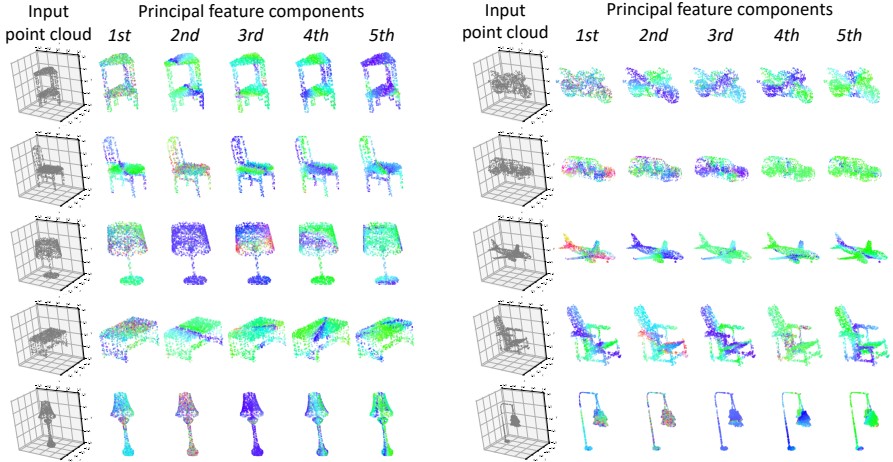

Figure 24: Visualization of principal feature components in PointNet learned on the ShapeNet dataset. Heatmaps were normalized using the second approach in Section D.

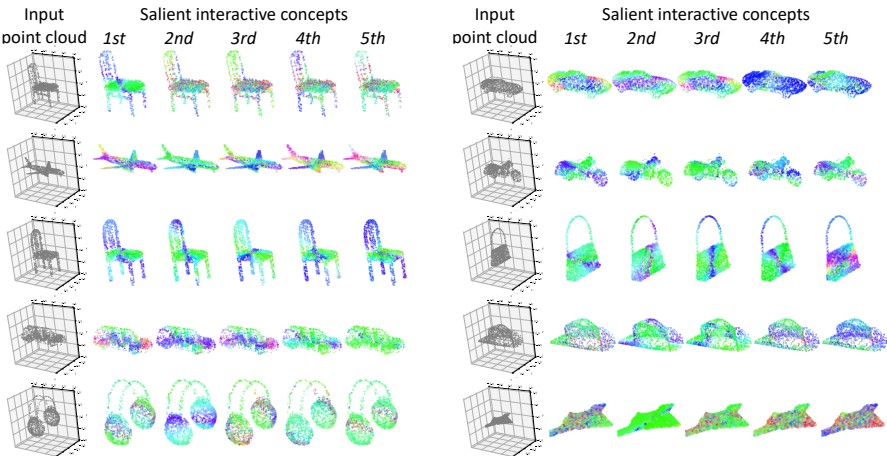

Figure 25: Visualization of salient interactions in PointNet learned on the ShapeNet dataset. Heatmaps were normalized using the second approach in Section D.

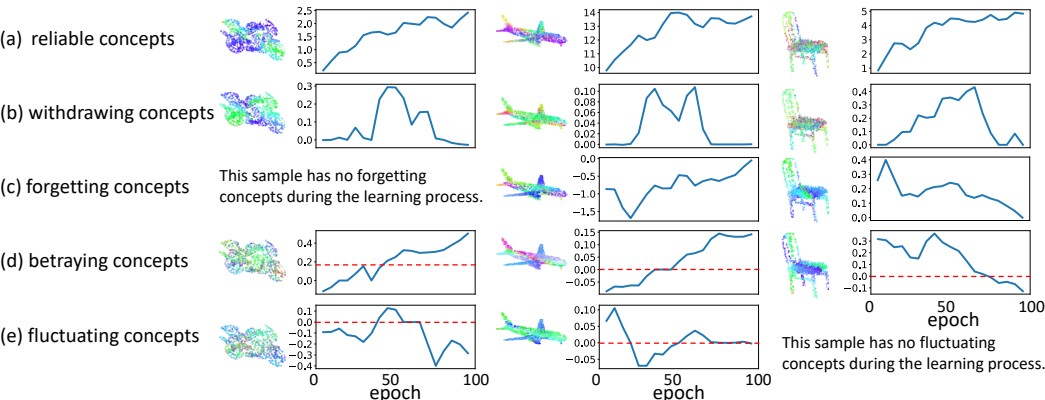

Figure 26: Curves of interactions in PointNet trained on the ShapeNet dataset.

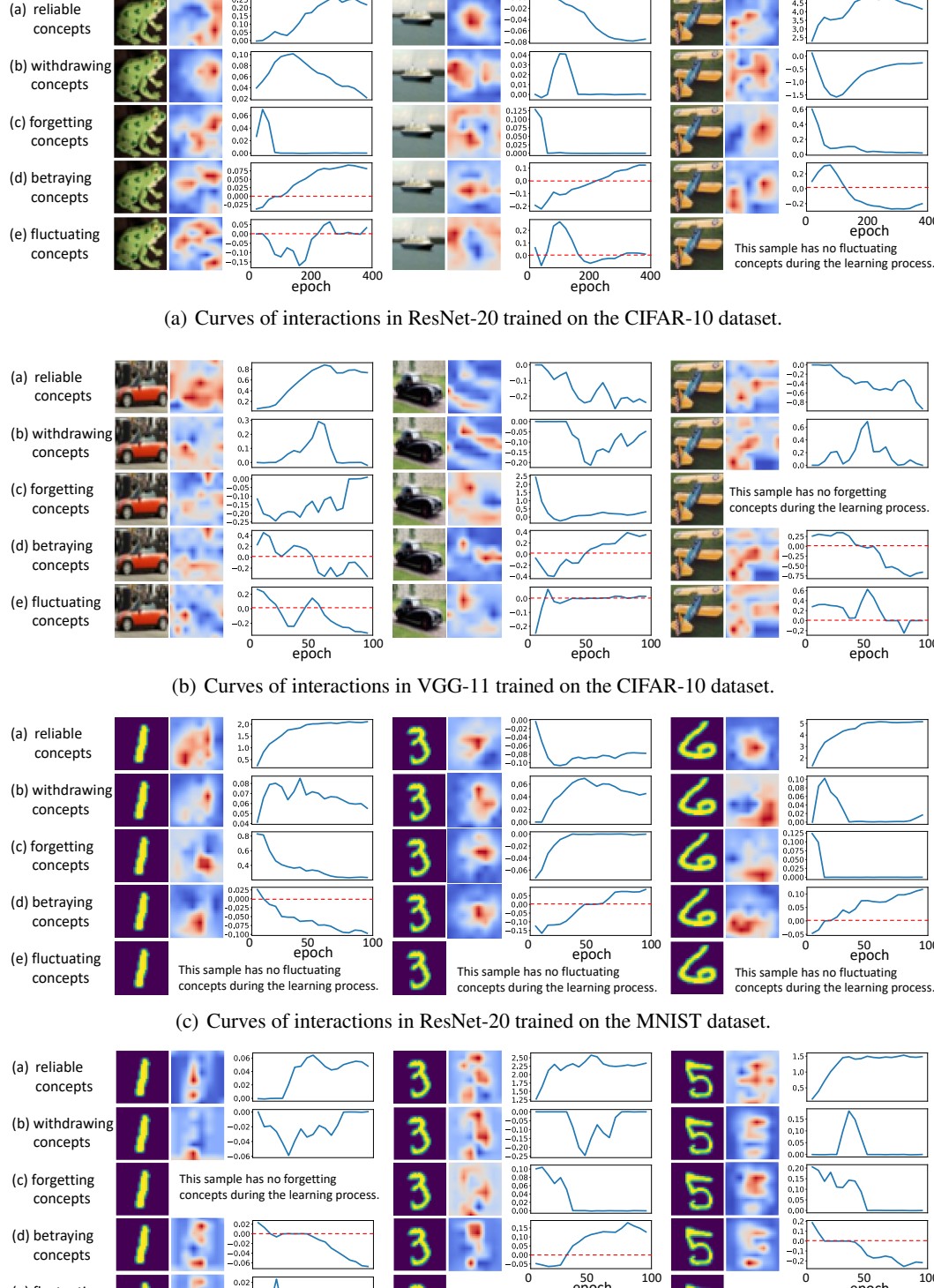

(a) Curves of interactions in ResNet-20 trained on the CIFAR-10 dataset.

(b) Curves of interactions in VGG-11 trained on the CIFAR-10 dataset.

(c) Curves of interactions in ResNet-20 trained on the MNIST dataset.

(d) Curves of interactions in VGG-11 trained on the MNIST dataset.

Figure 27: interactions can be categorized into five groups according to their curves during the learning process.

