# OpenReview forum: "Visualizing the Emergence of Primitive Interactions During the Training of DNNs"
_ICLR.cc/2024/Conference — ICLR 2024 Conference Withdrawn Submission_

### Official Review · Reviewer_8d78 · 2023-11-01

**Soundness:** 2 fair
**Presentation:** 3 good
**Contribution:** 2 fair
**Rating:** 3
**Confidence:** 4

**Summary:**

This paper builds upon past work that attempts to explain the behavior of DNNs through the lens of interactions of features (or groups of features). These interactions are characterized by (a) a hard subset of features, i.e. "all else being equal"-type interactions (b) their predictive strength, i.e. if the model sees only this subset, how far off is it.

In this work in particular, this is achieved not in input space but rather in a PCA decomposition of the latent space. Great care is taken to ensure that removing a feature maintains consistency within the model (e.g. by taking the average activation). The paper shows that (a) in simple problems few interaction hold most of the predictive power, and (b) these interactions can be given some semantic label.

**Strengths:**

The paper tackles an interesting problem and presents it well. The results are not too surprising given the literature on the topic, but the investigation presents an interesting angle on the topic.

**Weaknesses:**

The main weakness of this paper is that I'm not sure I've learned much by reading it, therefore I'm worried about its significance. There are two aspects to this. The first, which makes me also concerned for novelty, is that this paper mostly builds on prior work except for applying the methods on a PCA decomposition rather than on input patches.

Second, the buildup to the binning of features into 5 categories was interesting, almost captivating, but then the binning itself was quite underwhelming. Why these 5 categories? How are they formally defined? (As it is, it feels like a human looked at each curve and decided; Appendix E does not clarify the matter.) Do they have any predictive power?

What should be the main contribution of this paper feels like an afterthought. Here are some suggestions which I think could improve this work:
- Most importantly, formalize these 5 categories into formulas or very clear algorithms. Show that these are robust, distinct categories through clustering, PCA, t-sne, or somehow.
- Experiment with design choices, and validate that these categories reflect _known_ or expected behaviors of DNNs as the capacity/kind/architecture of a model changes. For example the authors posit that residual connections have some precise impact. This should be an experiment.
- Find the predictive power of these categories, for example, do models with more fluctuating interactions overfit more?

I'm hopeful for this work but I don't think it its current form this paper is a good contribution, accepting it now would feel like a missed opportunity for the authors to deliver something much more interesting and impactful.

**Questions:**

> Nevertheless, the number of salient interactions with considerable utilities is still significantly lower than the exponential number of all potential $2^{n+1}$ interactions

This does seem almost certainly true. That being said, the search for interesting subsets does still require a search over $O(2^n)$ interactions. The solution proposed in the paper to only look at the top-r (10?) PCs is a bit underwhelming, and feels like it would miss a very heavy tail of interesting interactions for any interesting dataset [1].

Figures 3 & 4 should really have a logarithmically scaled y-axis.
Generally, I'm not a fan of the figures, they are all pretty small and it is hard to discern what is going on. I'd suggest thinking of different ways to present this information, probably by overlaying these curves into one plot or presenting the information in a different way (via different quantities).

[1] interesting dataset, which, while I haven't commented on it above, I think is missing from this paper. I understand these experiments can be computationally demanding, but at the same time, MNIST can be solved to 93% accuracy with a linear classifier, CIFAR-10 to 86% with a 75k-parameter ResNet. These are not very "rich" datasets for which one would expect lots of interesting interactions.

---

### Official Review · Reviewer_UCkY · 2023-11-06

**Soundness:** 4 excellent
**Presentation:** 3 good
**Contribution:** 3 good
**Rating:** 6
**Confidence:** 3

**Summary:**

In this paper, the authors address the extraction and visualization of primitive interactions, based on the Harsanyi dividend, in the learning dynamics of deep neural networks. Estimating the relevant saliency regions of the input is known to be a hard problem, due to the exponential scaling of possible input variables subsets. The authors hence introduce a PCA estimation of the most relevant components in an intermediate hidden layer by collecting the activations coming from different epochs. This allows them to determine the Harsanyi dividend by composing different hidden feature projections, instead of masking input variables. With the experiments, it is shown that these interactions can be distinguished based on a taxonomy of 5 different temporal behaviors, shedding light on the kind of interactions different models are typically using for solving training tasks.

**Strengths:**

This paper follows the research line of explaining the inner workings of neural networks by using a generalization of Shapley values to regions. Previous works have assessed that the AND and OR Harsanyi interactions that arise in the neural network are sparse and are actually useful for visualizing the relevant input variables for the predictions. The major contribution of this work is presenting a way to estimate the interactions in a simpler manner by computing the principal components in the temporal ensemble of hidden activations. Doing this alleviates the computational cost of previous methods while highlighting different behaviors during training.

The proposed method is sound and attains good sparsity results in all preliminary analyses, showing it is relevant for evaluating the primitive interactions of the model. Moreover, the use of PCA analysis connects this formulation to other approaches in XAI that instead mine concepts in the hidden layers of DNNs (see the questions). This could be useful to reconcile two apparently different research directions in XAI.

I found writing is clear and the paper's claims are sound, including a good quantity of material in the supplementary. I suggest the authors also include their implementation code for reproducibility.

**Weaknesses:**

One aspect that has not been analyzed thoroughly in the paper is the actual consistency between the estimation of the primitive interactions at the input level and at the intermediate level. It somehow sidelined in the main text how the estimation of the interactions in an intermediate layer is a consistent estimator of the interactions at the input level.  Given that, passing to intermediate representations may hinder somehow the quality of the interactions that are obtained at the input: there is the risk that two different intermediate-level interactions will capture the same input-level interaction. An example of this can be noted by the pretty-quite similar activation maps in  Figure (b) for the 1st and 2nd salient interactions of the 0 digit. I guess that given this result more discussion on this point should be addressed.

Some other details are less clear from the text, which however do not constitute a major weakness in the presented material. One aspect is that the discussion on the five groups of interactions is quite short and lacks details about the results in Figure (10). It is said that the 100 maximally higher interactions were selected for each DNN, but are the results specific for one sample or averaged over the dataset? Also, in Figure 5 it is said that it is measured the "average error" by using the Top $\alpha$ components, but I understood that the y-axis measures the network output, whereas the shaded region is the approximation error. Is the blue line the ground truth value for the sorted $S$?

**Questions:**

Recent works in post-hoc concept-based explanations are addressing mining with PCA or SVD. The main difference with the proposed method here is that, instead of considering the time evolution, they take the principal components for all data representations belonging to a specific class, see [1,2,3]. Would it be the case that the same method would work also to estimate Harsanyi interactions?

[1] Ghorbani, Amirata, et al. "Towards automatic concept-based explanations." NeurIPS (2019).
[2] Zhang, Ruihan, et al. "Invertible concept-based explanations for CNN models with non-negative concept activation vectors."AAAI (2021)
[3] Fel et al. 2023, A Holistic Approach to Unifying Automatic Concept Extraction and Concept Importance Estimation, NeurIPS (2023)

---

### Official Review · Reviewer_kGww · 2023-11-07

**Soundness:** 2 fair
**Presentation:** 2 fair
**Contribution:** 2 fair
**Rating:** 3
**Confidence:** 4

**Summary:**

The paper visualizes the learning process for specific layers in specific DNNs on specific datasets by viewing and categorizing interactions between PCA features and categorizes these into different semantic groups based on how their utility scores evolve as training progresses.

**Strengths:**

**Originality**
- The paper extends work that shows that DNNs encode sparse interactions, which are considered "primitives", into further sparse interactions by taking the PCA of features extracted from an intermediate layer of certain DNN models training on certain datasets.
- The paper shows the approximation error by using lower-rank PCA features instead of a large number of raw features (e.g., patches on image data).
- The paper creates visualizations of the PCA-derived *principal feature components* and *salient interactions* using heatmaps for image data, gradient magnitudes for point cloud data, and Shapley values for language data.
- The paper categorizes the salient interactions encoded in the DNN into five semantically-meaningful groups based on the properties of these interactions as training progresses..

**Quality**
- Definitions of various quantities, such as the $i$-th principal feature component $f_i$ are rigorously defined and intuitively or rigorously justified, though thresholds, such as rank $r = 10$ and the $0.1$, or $90\%$ in the minimum ratio of the most salient interactions, are some examples of seemingly ad-hoc chosen values.

**Clarity**
- For a relatively dense exposition of preliminaries, the paper does an passable job at clarifying these preliminaries though I believe it could benefit greatly from elucidating its decision-making process of using certain values and of certain claims that it makes, which I'll detail in Weaknesses.
- Sections 3 and 4 comprise the bulk of the paper and are adequately comprehensive at this stage of the review process.
- The paper justifies the use of certain models and datasets by aligning some of them with those used in previous work that it compares against.

**Significance**
- I think the categorization of salient feature types could inspire others to continue looking at semantically-meaningful features and ideally marry this intuitive quantitative categorization with reliability and rigor.

**Weaknesses:**

- As far as I see, there is no mention of limitations of this work, let alone a Limitations section. No work is perfect, and every work should include a Limitations section so that, only two reasons given here for concision, (1) readers are quickly aware of cases in which this work applies and in which it doesn't and (2) readers have confidence that the paper is at least somewhat cognizant of (1). I'm unsure whether this is in the Appendix or Supplementary Material.
- Very limited Related Works section. A large section of related works that is relevant is "sparsity in neural networks," and this could be broken down into multiple relevant subsections, such as "sparsity over training progress", "sparsity with respect to {eigenvalues, spectral norms, Hessian properties [1], etc.}"
- Limited rigor in original (at least original as far as I know, such as the categorization of salient features) concepts.
  - What quantitative rigor justifies the categorization of a feature into one of the 5 mentioned categories?
  - Is there some sort of goodness of fit test or statistical hypothesis test or principled approach for assigning a feature to a category?
  - What if the training epochs were extended and the utility trended in a way that changed categorization?
    - What was the stopping criteria for training?
  - Was any analysis done for the reliability of assigning features to categories?
- Unclear in several aspects. Some include
  - Why use only one layer for each of the DNNs? How was this layer selected? How would results changing using a different intermediate layer?
  - Why use the threshold values for rank, approximation error for salient feature count, the number of training epochs used, among others?
  - Are the results in Figure 5a, 5b, and 5c each for one "sample", "sentence", and "image" in the single DNN model and single dataset listed?
- Do Figures X and Y show results for randomly sampled images? Since it's impossible to confirm whether this was actually the case, are there examples that do not align with these results, or even contradict these results? Is there analysis as to why?
- The novelty of using PCA to reduce interaction count seems incremental and the significance of the paper results is unclear to me. Using PCA to reduce the interaction count seems intuitive, as PCA aims to retain the maximum information in the data with the reduced dimensionality chosen, assuming certain assumptions are met. How well are the assumptions met?

[1] Dombrowski, Ann-Kathrin, Christopher J. Anders, Klaus-Robert Müller, and Pan Kessel. "Towards robust explanations for deep neural networks." Pattern Recognition 121 (2022): 108194.

**Questions:**

The main questions I have for the paper are included in the Weaknesses section, and some implied questions are in the Strengths section. I believe addressing these questions will result in a stronger paper.

**Details Of Ethics Concerns:**

I do not have ethics concerns for this work.